# An Asymmetrical 19-Level Inverter with a Reduced Number of Switches and Capacitors

Farzad Sagvand, Jafar Siahbalaee * and Amangaldi Koochaki

Department of Electrical Engineering, Aliabad Katoul Branch, Islamic Azad University,
Aliabad Katoul 4941793451, Iran
* Correspondence: j.siahbalaee@aliabadiau.ac.ir

**Abstract:** Multilevel inverters are able to provide loads with voltages of high power quality using several DC sources, capacitors, switches, and diodes in their structures. However, the usage of the higher number of semiconductor devices (switches and diodes) and capacitors causes an increase in losses and costs and decreases their efficiency. Thus, lowering the number of switches and capacitors is a challenging issue in designing a multilevel inverter. In this paper, an asymmetrical multilevel inverter is proposed that produces 19-level output voltages. The circuit is composed of nine switches, six diodes, two capacitors, and two isolated DC sources. In comparison with other topologies, the most important advantage of the introduced 19-level topology is the usage of a lower number of switches and capacitors, which leads to a decrease in the number of gate drivers and the total volume of the system. During the charging process, capacitors never connect to each other in series, i.e., they are self-balancing and do not require the extra circuits. The proposed topology offers a total harmonic distortion (THD) of 7.4% in the output voltage, which is less than 8%, complying with the IEEE standards. The performance of the topology is validated under various load conditioning through an experimental setup in the laboratory.

**Keywords:** multilevel inverter; asymmetrical converter; reduced device count; multi-carrier pulse width modulation





## 1. Introduction

The multilevel inverter (MLI) is one type of power electronic converter used for medium voltages and high-power applications. Compared to the two-level inverters, it is able to provide voltages with low total harmonic distortion (THD), low voltage rating on the semiconductor devices, and high power quality for the load. In MLIs, the switching frequency of switches can be reduced, which leads to a decrease in the power losses of the semiconductor devices and an increase in the efficiency of the inverter [1–3]. There are three basic topologies of MLIs: Neutral Point Clamped (NPC) [4], Flying Capacitor (FC) [5], and Cascade H-Bridge (CHB) [6,7]. One important advantage of NPC and FC topologies is that they use just one isolated DC source in their structures. However, to provide a higher number of voltage levels in the output, they need more switches ($N_{sw}$), diodes ($N_d$), and capacitors ($N_C$). For instance, to provide a load with 5-level voltages, NPC and FC topologies require the numbers ($N_{sw}$, $N_d$, and $N_C$), respectively, to be equal to (8, 6, 4) and (10, 0, 8). Another drawback of the NPC and FC topologies is neutral point balancing, which involves using an extra circuit. To overcome these issues, numerous types of research have been conducted; nonetheless, the proposed designs still suffer from large numbers of devices in their structures [8,9]. The CHB-based topologies of MLIs have been developed due to the modularity of their structures, fewer capacitors, and simple controllability. They have been widely used for photovoltaic systems, rechargeable batteries, electrical vehicles, reactive power compensators, and so forth [10–12]. The CHB topology can be configured in asymmetrical mode (inequality

of input DC sources) to achieve more voltage levels. The main disadvantage of CHB topologies is the usage of several isolated DC sources in their structures. Recently, the proposed switched-source (SS)- and switched-capacitor (SC)-based topologies have reduced the number of devices. The SC configurations can play the role of the boost converter for photovoltaic applications [13]. Generally, in SS and SC structures, to achieve a higher number of voltage levels, and lower THD on the load, the number of DC sources and capacitors should be increased [14,15]. However, compared to the NPC and FC topologies, they benefit from a lower number of switches and diodes [16,17]. In [18], a multilevel inverter topology was introduced with a very low number of switches. However, the total standing voltage (TSV) of the proposed inverter was equal to 9. In [19], a new multilevel inverter structure was presented by Babaei with 10 switches and 3 DC sources to obtain the 13-level output voltage. The introduced multilevel inverter topology in [20] had 14 switches and 2 capacitors. This topology had a high number of switches but the TSV was 5.33. From the above discussion, it can be concluded that by decreasing the number of switches, the TSV increases and vice versa. Hence, a multilevel inverter topology can be designed with a low number of switches and satisfy the value of TSV. This paper presents an asymmetrical 19-level hybrid switched source-capacitor inverter that has the ability to provide the merits of SS and SC topologies. The proposed structure consists of nine switches, six diodes, two capacitors, and two isolated DC sources, which act as the boost converter with a voltage gain of 2.25. The number of switches, gate drivers, and capacitors in the proposed 19-level topology is very low and comparable with the recently suggested topologies. However, the TSV of the topology is 7.2, but this is satisfied in comparison with other structures such as [18]. Multi-carrier pulse width modulation (MC-PWM) is chosen as the switching strategy because it features the multilevel inverter as a controllable apparatus.

The paper is organized as follows. In Section 2, the proposed topology is presented, accompanied by a description of the performance of the switches. The generalized topology is illustrated in Section 3. Sections 4–6, respectively, present a comparison of the proposed multilevel inverter with other topologies, detail the MC-PWM scheme, and conduct calculation of the losses and efficiency, respectively. To verify the performance of the inverter, the results of the laboratory tests are given in Section 7. Finally, Section 8 concludes the paper.

## 2. Proposed 19-Level Inverter

Figure 1 shows the proposed asymmetrical 19-level inverter. It consists of two units, nine switches, two capacitors, six diodes, and two DC sources. Switch $S_5$ is only applied to charge capacitors $C_1$ and $C_2$. In addition, pairs of switches ($S_1$, $S_3$) and ($S_2$, $S_4$) are assigned to deliver the voltage of the DC sources and capacitors to the load, respectively. The steps of magnitude of the DC sources are introduced as $u_1$ = 3 V and $u_2$ = V. In other words, if we choose V = 20 v, the first and second input DC bus sources are set at $u_1$ = 60 v and $u_2$ = 20 v, respectively. Because the maximum voltage level on the load is 9 V, the magnitude of output voltage of the inverter is 180 V. Assuming the values of DC sources as $u_1$ = 3 V and $u_2$ = V, supposedly in asymmetrical conditions, capacitors $C_1$ and $C_2$ would be charged up to 4 V and 1 V, respectively. Thus, the four available voltage sources V, V, 3 V, and 4 V can give rise to a 19-level voltage on the output of the inverter. It should be noted that switch $S_5$ must have no body diode, due to the unidirectional flow of the current responsible for charging the capacitors. The switching states of the proposed 19-level inverter are listed in Table 1. In order to create more frequent capacitor charging, the zero voltage level is bisected into two parts (0+, 0−) according to states 10 and 11. As the capacitors are charged by only one switch ($S_5$) simultaneously, there are only four switching states through which the capacitors are charged. Hence, the discharging time of the capacitors is expected to be longer than the charging time. To overcome the issue, each unit must have a separate switch to charge its capacitor, which leads to increasing the number of switches. As a result, in order to avoid this effect, there

must be only one common switch ($S_5$) to charge the capacitors. However, this issue, again, leads to simultaneous charging of the capacitors, which, in turn, causes the discharging time of capacitors to be larger than their charging time. Figure 2 shows the circuit configuration and paths of the current flow during the positive and zero levels of operation.

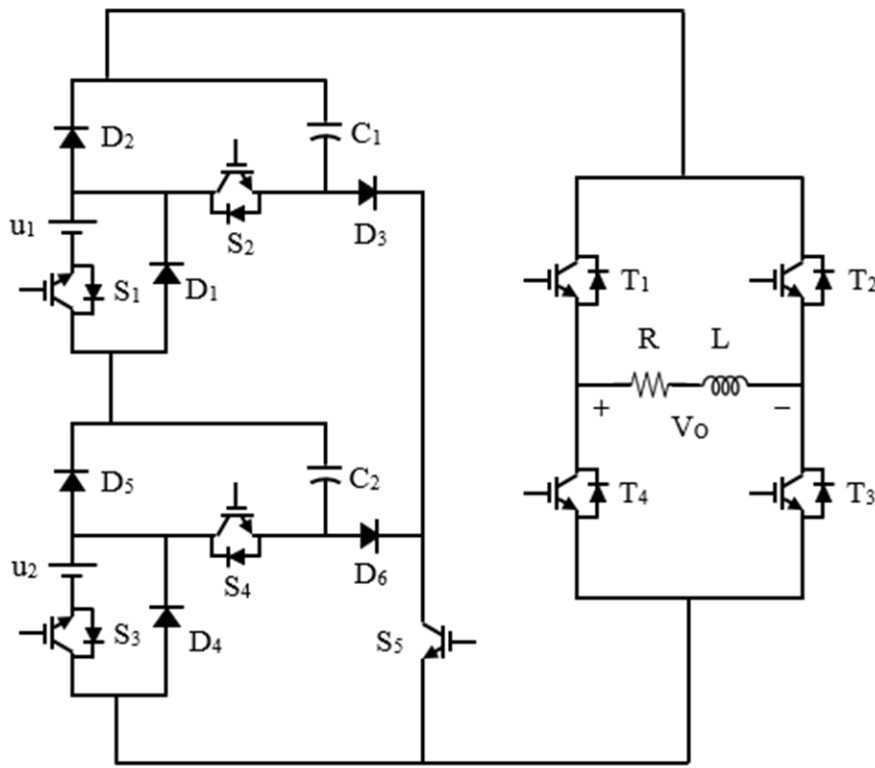

**Figure 1.** Proposed 19-level inverter with $u_1 = 3$ V and $u_2 = V$.

**Table 1.** Switching states of the proposed 19-level inverter (symbols C, D, and W indicate charge, discharge, and without change, respectively).

| States | $S_1S_2S_3S_4S_5$ | $T_1T_2T_3T_4$ | $C_1C_2$ | $V_{out}$ |
|---|---|---|---|---|
| 1 | 11110 | 1010 | D-D | +9 V |
| 2 | 11100 | 1010 | D-W | +8 V |
| 3 | 11000 | 1010 | D-W | +7 V |
| 4 | 01110 | 1010 | D-D | +6 V |
| 5 | 01100 | 1010 | D-W | +5 V |
| 6 | 10101 | 1010 | C-C | +4 V |
| 7 | 10100 | 1010 | W-W | +3 V |
| 8 | 00110 | 1010 | W-D | +2 V |
| 9 | 00100 | 1010 | W-W | +V |
| 10 | 10101 | 1100 | C-C | 0+ |
| 11 | 10101 | 1100 | C-C | 0− |
| 12 | 00100 | 0101 | W-W | −V |
| 13 | 00110 | 0101 | W-D | −2 V |
| 14 | 10100 | 0101 | W-W | −3 V |
| 15 | 10101 | 0101 | C-C | −4 V |
| 16 | 01100 | 0101 | D-W | −5 V |
| 17 | 01110 | 0101 | D-D | −6 V |
| 18 | 11000 | 0101 | D-W | −7 V |
| 19 | 11100 | 0101 | D-W | −8 V |
| 20 | 11110 | 0101 | D-D | −9 V |

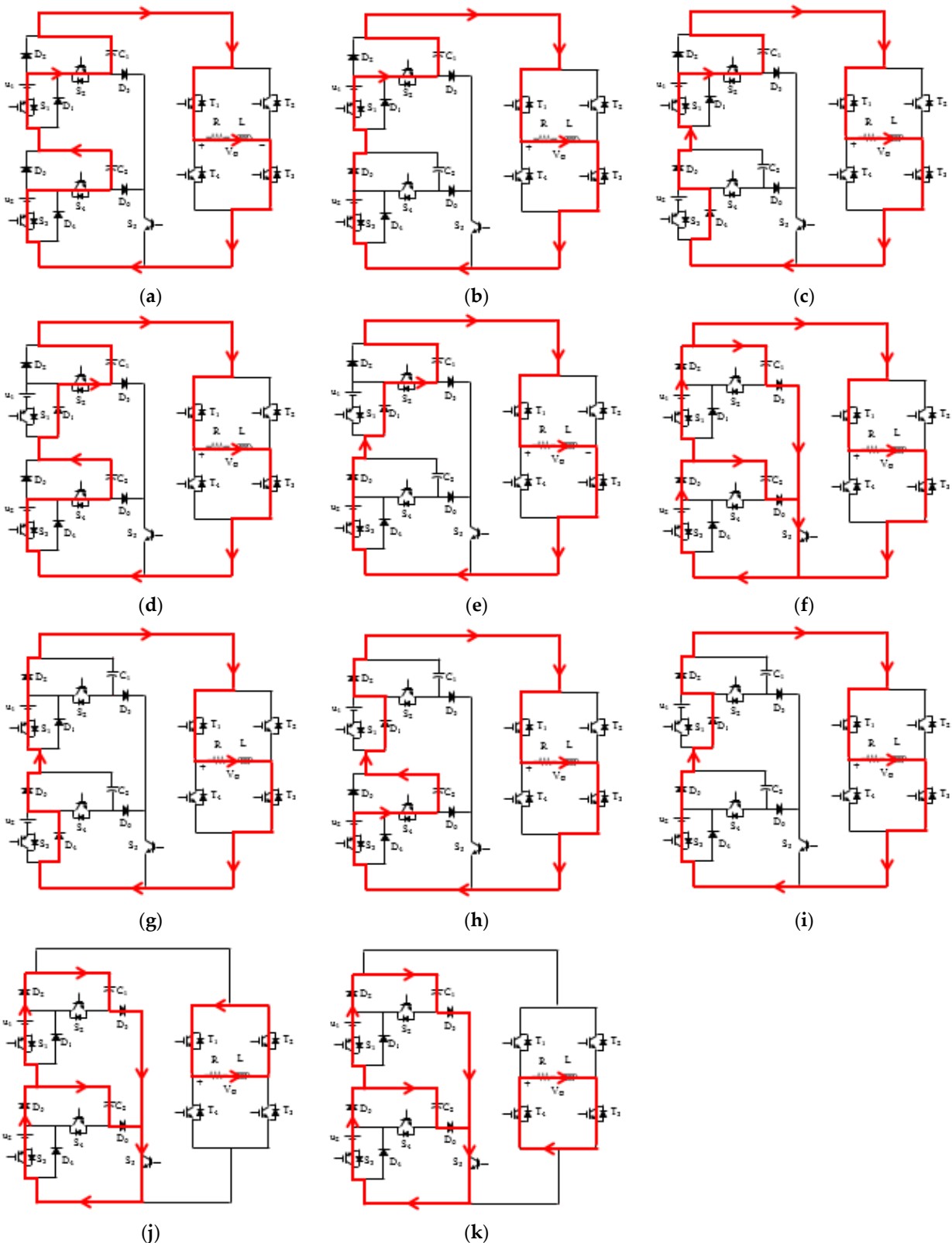

**Figure 2.** Schematic of switching states for generating different positive and zero levels on the load with $u_1 = 3$ V and $u_2 = 1$ V (see Table 1): (**a**) state 1 for $V_O = +9$ V, (**b**) state 2 for $V_O = +8$ V, (**c**) state 3 for $V_O = +7$ V, (**d**) state 4 for $V_O = +6$ V, (**e**) state 5 for $V_O = +5$ V, (**f**) state 6 for $V_O = +4$ V, (**g**) state 7 for $V_O = +3$ V, (**h**) state 8 for $V_O = +2$ V, (**i**) state 9 for $V_O = +$V, (**j**) state 10 for $V_O = 0+$, and (**k**) state 11 for $V_O = 0-$.

For instance, in Figure 2a, which describes state 1 of Table 1, when switches $S_1$, $S_2$, $S_3$, and $S_4$ are turned on, diodes $D_1$, $D_2$, $D_4$, and $D_5$ are in reversed bias. In these conditions, the current passes through devices $S_3$, $u_2$, $S_4$, $C_2$, $S_1$, $u_1$, $S_2$, $C_1$, $T_1$, load, and $T_3$, which, in turn, produces a +9 V level across the output. Other schematics can be analyzed in the same way.

### 3. The Proposed Generalized Multilevel Inverter

If devices $u_1$, $S_1$, $S_2$, $D_1$, $D_2$, and $C_1$ form a unit, the structure of the proposed 19-level inverter, shown in Figure 1, would consist of two units. As shown in Figure 3, in order to achieve more voltage levels, more units can be utilized. In such cases, based on the values of DC sources, several asymmetrical modes can be defined for the proposed multilevel inverter. The first, second, and third asymmetrical modes are assumed to be (V, 2 V, 3 V, ... , nV), (V, 3 V, 5 V, ... , (2n−1) V), and (V, 3 V, 10 V, 34 V, ... ), respectively. Although the third asymmetrical mode produces a higher number of voltage levels than the first and second ones, in this case, the total standing voltage (TSV) increases in some switches, which necessitates the usage of switches with higher rated voltage; this, in turn, causes higher costs. For the case of n units, the number of devices needed for the three aforementioned asymmetrical modes can be written as

$$\begin{cases} N_C = n, & n \in N \\ N_{sw} = 2n + 5, & n \in N \\ N_d = 3n, & n \in N \end{cases} \tag{1}$$

where $N_C$, $N_{sw}$, and $N_d$ are the number of capacitors, switches, and diodes, respectively. According to Figure 3, in the third asymmetrical mode, values of the DC-source and capacitor voltages of unit n can be obtained as

$$\begin{cases} E_n = \sum\limits_{i=1}^{n-1} E_i + \sum\limits_{i=1}^{n-1} E_{ci} + 1 \\ E_{cn} = \sum\limits_{i=1}^{n} E_i \end{cases} \tag{2}$$

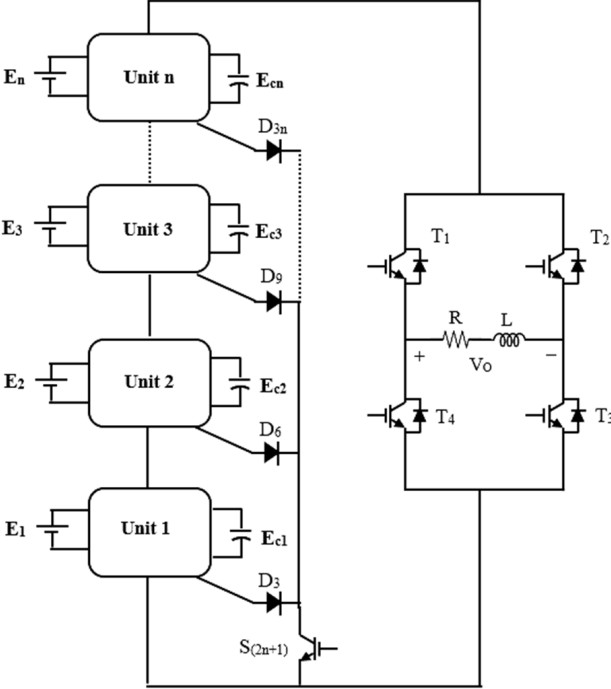

**Figure 3.** The proposed generalized multilevel inverter with n units.

The relationship between the number of required devices and the voltage levels is listed in Table 2. For example, to reach the 33-level voltages across the load, we need 11 switches, 9 diodes, 3 units, 3 capacitors, and 3 DC sources; these numbers account for only the first asymmetrical mode.

**Table 2.** The number of necessary devices in the proposed multilevel inverter for previously defined asymmetrical modes ($n_{level}$ is the number of voltage levels).

| $N_{Level}$ | First Symmetrical Mode | | | Second Asymmetrical Mode | | | Third Asymmetrical Mode | | |
|---|---|---|---|---|---|---|---|---|---|
| | $N_{sw}$ | $N_d$ | $N_C$ | $N_{sw}$ | $N_d$ | $N_C$ | $N_{sw}$ | $N_d$ | $N_C$ |
| 5 | 7 | 3 | 1 | 7 | 3 | 1 | 7 | 3 | 1 |
| 15 | 9 | 6 | 2 | – | – | – | – | – | – |
| 19 | – | – | – | 9 | 6 | 2 | 9 | 6 | 2 |
| 33 | 11 | 9 | 3 | – | – | – | – | – | – |
| 47 | – | – | – | 11 | 9 | 3 | – | – | – |
| 61 | 13 | 12 | 4 | – | – | – | – | – | – |
| 67 | – | – | – | – | – | – | 11 | 9 | 3 |
| 93 | – | – | – | 13 | 12 | 4 | – | – | – |
| 101 | 15 | 15 | 5 | – | – | – | – | – | – |
| 161 | – | – | – | 15 | 15 | 5 | – | – | – |
| 155 | 17 | 18 | 6 | – | – | – | – | – | – |
| 231 | – | – | – | – | – | – | 13 | 12 | 4 |
| 255 | – | – | – | 17 | 18 | 6 | – | – | – |

## 4. Comparison of the Proposed Multilevel Inverter with Other Topologies

The suggested multilevel inverter can be weighed up against other topologies in terms of the number of required switches and capacitors. Here, the NPC, FC, CHB, Prabaharan [18], Babaei [19], Wang [20], Barzegarkhoo [21], Samizadeh [22], and Roy [23] structures are chosen to be compared with the proposed configuration, as demonstrated in Figure 4.

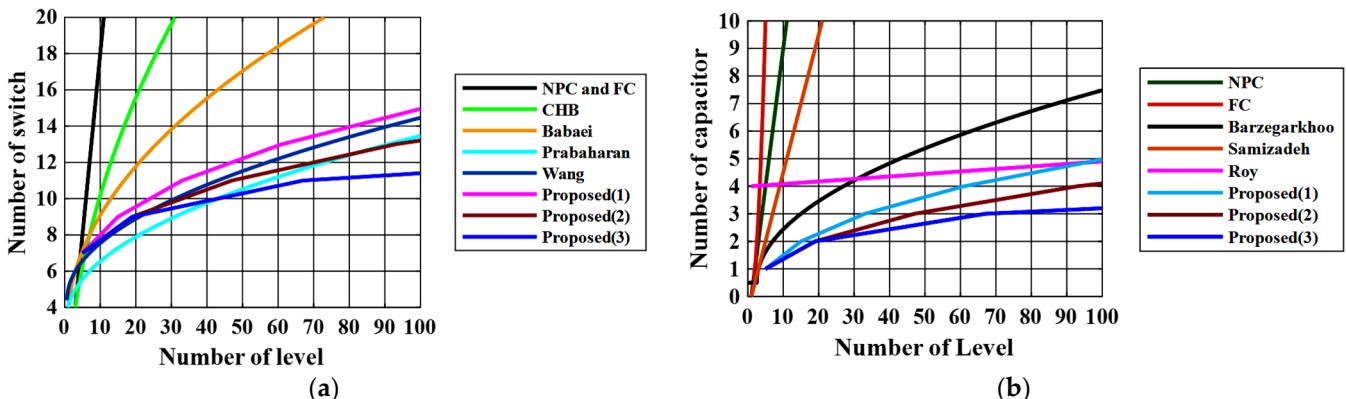

**Figure 4.** Comparison of the proposed multilevel inverter with other topologies. (**a**) Number of switches vs. number of levels; (**b**) number of capacitors vs. number of levels. The proposed (1), (2), and (3) correspond to the first, second, and third asymmetrical modes.

According to Figure 4a, the Prabaharan topology has the lowest number of switches as the number of voltage levels at the output are less than 84 (in point M), but when the number of voltage levels exceeds 84, this topology is outperformed by the proposed structure; that is, the proposed structure offers the minimum number of required switches

among all the designs. However, this phenomenon occurs in the second asymmetrical mode, i.e., V, 3 V, 5 V, . . . , (2n−1) V. On the other hand, as illustrated in Figure 4b, the suggested topology needs the lowest number of capacitors, both in the first and second asymmetrical modes. Therefore, as a result of the comparisons, the proposed inverter is able to produce a high number of voltage levels for the load by the low number of switches and capacitors. Nonetheless, more output voltage levels can be achieved at the cost of higher TSVs.

Table 3 shows the comparison of the proposed configuration with other topologies. According to this table, the 19-level proposed inverter has the lower number of switches and gate drivers than other conventional structures. Moreover, the voltage gain of the introduced topology is 2.25, which is more than the other studied topologies. However, it seems that the TSV of the proposed inverter is higher than those of the other topologies [18]. This is because the introduced multilevel inverter works at the asymmetrical mode with $u_1$ = 3 V and $u_2$ = V, which leads to an increase in the stress voltage of semiconductor devices.

**Table 3.** Comparison of proposed 19-level inverter structure with other topologies.

| Topology | $N_L$ | $N_{sw}$ | $N_d$ | $N_C$ | $N_{gd}$ | $V_G$ | $TSV^{pu}$ |
|----------|-------|----------|-------|-------|----------|-------|------------|
| [18] | 13 | 7 | 3 | 0 | 7 | 2.16 | 9 |
| [20] | 13 | 14 | 0 | 2 | 11 | 2 | 5.33 |
| [21] | 17 | 10 | 2 | 2 | 10 | 2 | 5.5 |
| [22] | 17 | 10 | 2 | 2 | 10 | 2 | 5.5 |
| [23] | 13 | 11 | 1 | 1 | 10 | 1.5 | 6.3 |
| [24] | 13 | 18 | 0 | 2 | 15 | 2 | 5 |
| [25] | 17 | 18 | 2 | 4 | 14 | 2 | 6 |
| [26] | 19 | 12 | 6 | 4 | 12 | 2.2 | 5.8 |
| [27] | 19 | 12 | 1 | 2 | 10 | 1.8 | 6.66 |
| Pro. | 19 | 9 | 6 | 2 | 9 | 2.25 | 7.2 |

## 5. Multi-Carrier Pulse Width Modulation Technique

According to Figure 5, in the MC-PWM switching technique, in order to produce the 19-level voltage at the MLI output, we need one sinusoidal 50 Hz reference signal with amplitude $A_{ref}$ and frequency $f_{ref}$, 18 triangular carrier signals, and 20 switching states. As shown in Figure 5, all of the carrier signals have the same amplitude ($A_t$) and frequency ($f_{sw}$), but they are shifted up and down relative to each other based on the applied $V_{DC}$.

According to this figure, in the positive half-cycle, the carrier signals $car_1$, $car_2$, $car_3$, . . . , and $car_9$ are responsible for the generation of the positive voltage levels, which provide the switching pulses for all the switches except $T_2$ and $T_4$. Similarly, in the negative half-cycle, the carrier signals $car_{10}$, $car_{11}$, $car_{12}$, . . . , and $car_{18}$ are responsible for the generation of the negative voltage levels, which provide the switching pulses for all the switches except $T_1$ and $T_3$.

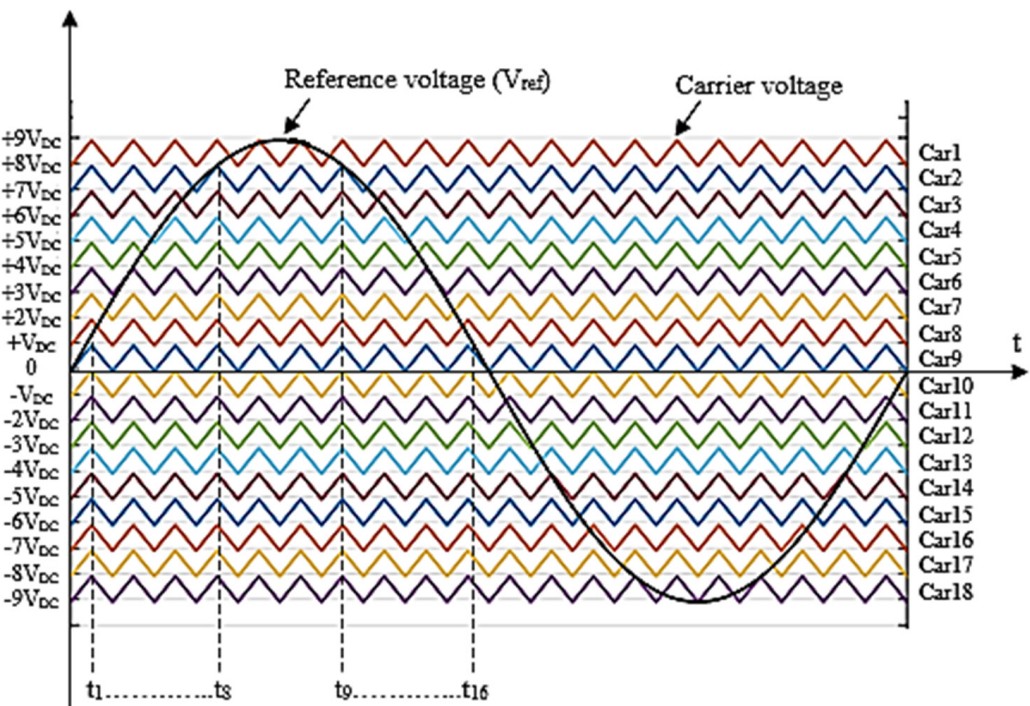

**Figure 5.** The MC-PWM technique for producing of the 19-level voltage at the proposed MLI output.

## 6. Calculation of Losses and Efficiency

In the proposed MLI, the losses consist of three parts: the capacitor charging losses ($P_{loss, cap}$), the switching losses ($P_{sw}$), and the conducting losses ($P_{cond}$).

### 6.1. The Capacitor Charging Losses

The capacitor charging losses are divided into the capacitor ripple losses ($P_{c, ripple}$) and the conduction losses of the capacitor ($P_{CC}$). The capacitor ripple losses, which occur due to voltage differences between the dc input and the voltage across the capacitors, are obtained by [24–26]:

$$P_{c,ripple} = \frac{f_{ref}}{2} \sum_{k=1}^{2} C_k \Delta V_{Ck}^{2} \tag{3}$$

where $\Delta V_{Ck}$ is the voltage ripple of $k$th capacitor, which can be written as

$$\Delta V_{Ck} = \frac{1}{C_k} \int_{t_{a,k}}^{t_{b,k}} i_{ck}(t) \cdot dt \tag{4}$$

where $i_{Ck}$ is the current flowing through the $k$th capacitor. In addition, the conduction losses of the capacitor, which are created by the internal resistance of capacitors ($R_C$), can be described as

$$P_{CC} = \left(\frac{2\pi f_{ref}}{\pi}\right) \sum_{k=1}^{2} \int_{t_{a,i}}^{t_{b,i}} R_C i_{ck}^{2} \cdot dt \tag{5}$$

Finally, the capacitor charging losses are calculated as

$$P_{loss,cap} = P_{C,ripple} + P_{CC} \tag{6}$$

### 6.2. The Switching Losses

The switching losses are rooted in the existing delay between changing the states of the switch from on to off and vice versa. These losses, present in switches and diodes, are the reason that creates the so-called switching losses in the proposed MLI. Therefore, a

faster switch or diode in terms of recovery time evidently has lower switching losses. The switching losses during the ON ($P_{sw,on}$) and OFF ($P_{sw,off}$) states of a typical switch can be calculated by (7) and (8), respectively [26]:

$$P_{sw,on} = \frac{f_s \cdot V_{off} \cdot I_{on} \cdot t_{on}}{6} \tag{7}$$

$$P_{sw,off} = \frac{f_s \cdot V_{off} \cdot I_{on} \cdot t_{off}}{6} \tag{8}$$

where $t_{on}$ and $t_{off}$ are the time intervals during which the switch turns on and turns off, respectively, $f_s$ is the switching frequency, $V_{off}$ is the voltage rating of the switch, and $I_{on}$ is the average load current. Similarly, the switching losses of a diode can be calculated as

$$P_{sw,D} = \frac{f_s \cdot V_{RM} \cdot I_{RM} \cdot t_B}{6} \tag{9}$$

where $V_{RM}$ and $I_{RM}$ are the maximum voltage and current of reverse recovery, respectively, and $t_B$ is the delay time of the reverse current. The total switching losses, then, can be formulated as follows:

$$P_{sw,total} = \sum_{i=1}^{N_{sw}} \left( \sum_{j=1}^{N_{on}} \left( P_{sw,on,ij} \right) + \sum_{j=1}^{N_{off}} P_{sw,off,ij} \right) + \sum_{k=1}^{N_d} \left( \sum_{h=1}^{N_{off}} \left( P_{sw,D,kh} \right) \right) \tag{10}$$

where $N_{on}$ and $N_{off}$ are, respectively, the numbers of ON and OFF states of the switches and diodes during a complete fundamental cycle (1/Ts).

*6.3. The Conducting Losses*

Two key factors are the main causes of the conducting losses. One is the internal resistance of each semiconductor device and the other is the voltage of their ON state. These together create the voltage drop on the semiconductor devices. The conducting losses on a switch ($P_{cond,sw}$) and diode ($P_{cond,D}$) can be written as [27]

$$P_{cond,sw} = V_{on,sw} \cdot I_{sw,ave} + R_{on,sw} \cdot I_{sw,rms}{}^2 \tag{11}$$

$$P_{cond,D} = V_{on,D} \cdot I_{D,ave} + R_{on,D} \cdot I_{D,rms}{}^2 \tag{12}$$

where $V_{on}$ and $R_{on}$ are the voltage and resistance of the switch and diode during the ON state, respectively. In addition, $I_{rms}$ and $I_{ave}$ are the RMS and average current of the semiconductors, respectively. In MLIs, each voltage level creates a conducting loss. For example, according to Figure 2, in steps +(8, 9) $V_{DC}$, the conduction losses can be obtained using [28–30]

$$P_{cond,(+9V)} = \left(6V_{on,sw} \cdot I_{load,ave} + 6R_{on,sw} \cdot I_{load,rms}{}^2\right) + \left(0 \times V_{on,D} \cdot I_{load,ave} + 0 \times R_{on,D} \cdot I_{load,rms}{}^2\right) \tag{13}$$

$$P_{cond,(+8V)} = \left(5V_{on,sw} \cdot I_{load,ave} + 5R_{on,sw} \cdot I_{load,rms}{}^2\right) + \left(1 \times V_{on,D} \cdot I_{load,ave} + 1 \times R_{on,D} \cdot I_{load,rms}{}^2\right) \tag{14}$$

For the purpose of clarification, notice that in step +8 V, according to Figure 2b, there are five switches and one diode in the current commutation path. Therefore, the equation for the corresponding conduction losses of this step must account for five switches and one diode. The calculation of the conducting losses for the other steps follows a similar procedure. The total conduction losses, then, are the sum of losses of all the steps, which is

$$P_{cond,total} = P_{cond,(+9V)} + P_{cond,(+8V)} + P_{cond,(+7V)} + \ldots + P_{cond,(-7V)} + P_{cond,(-8V)} + P_{cond,(-9V)} \tag{15}$$

Accordingly, the efficiency of the proposed 19-level topology can be calculated as

$$\eta = \left( \frac{P_{out}}{P_{out} + P_{loss}} \right) \times 100 = \left( \frac{\frac{\left(V_{out(rms)}\right)^2}{R_{load}}}{\frac{\left(V_{out(rms)}\right)^2}{R_{load}} + P_{loss,cap} + P_{sw,total} + P_{cond,total}} \right) \times 100 \quad (16)$$

Figure 6 illustrates the efficiency curve as load power changes. As can be seen from the figure, in lower powers, the efficiency is low due to the small amplitude of the load current. However, as the load power rises up, the efficiency increases too until it reaches the maximum value, which is equal to 93.6% for the proposed 19-level inverter.

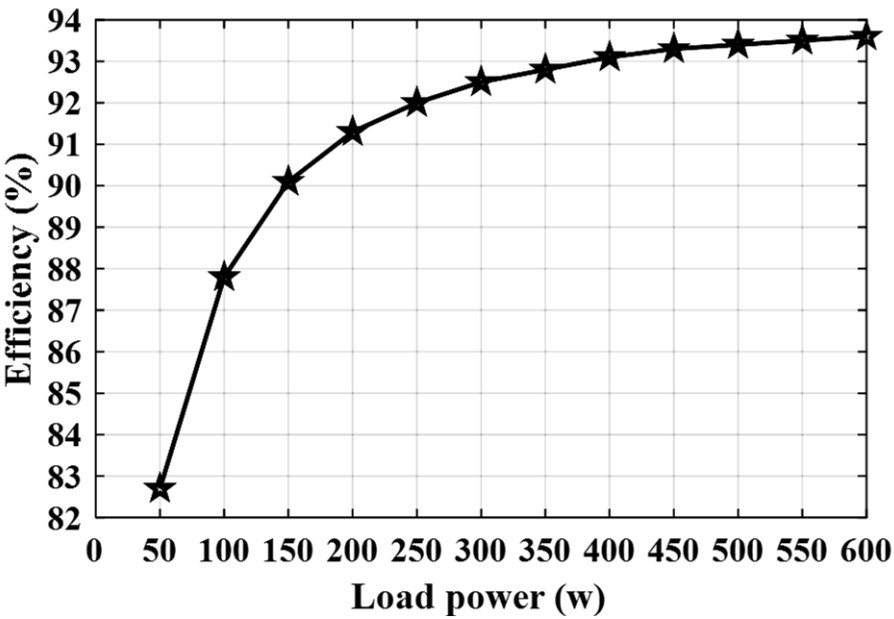

**Figure 6.** Efficiency curve vs. load power.

## 7. Simulation and Experimental Results

### 7.1. Simulation Results

In order to validate the performance of the proposed 19-level inverter, several simulations are carried out using SIMULINK/MATLAB. The DC bus voltages $u_1$ and $u_2$ are set at 60 V and 20 V, respectively. The switches and diodes are modeled as similar to semiconductor devices applied to the laboratory. For example, the inner resistance and voltage drop of switches and diodes are set at (0.07 $\Omega$, 1.2 v) and (0.065 $\Omega$, 0.85 v), respectively. Moreover, the values of both capacitors $C_1$ and $C_2$ are arranged as 4700 $\mu$F. Figure 7 shows the simulation results for output voltage, output current, and capacitor voltages. As shown in Figure 7, the output voltage forms a 19-level waveform under the MC-PWM switching strategy. In addition, the capacitor voltages of $C_1$ and $C_2$ are close to 80 V and 20 V, respectively, which comply with the values of ($u_1 + u_2$) and $u_2$, respectively. Hence, the capacitors are charged up to 4 V and 1 V according to the presented discussion in Section 2. Figure 7b shows the results with the increase in the load from Z = 300 $\Omega$ to Z = 150 $\Omega$ at t = 0.05 s. It can be seen from this figure that the current magnitude increases as 2 times. In Figure 7c, the load changes from pure resistive to inductive-resistive conditions. For this reason, the current wave is closed to sinusoidal form. In Figure 7d,e, the modulation index decreases from m = 1 to m = 0.5. In these conditions, the output voltage varies from a 19-level to 11-level form. This is because with m = 0.5, the reference voltage shown in Figure 5 decreases as much as half its previous value. Thus, in this case, the reference voltage is compared to 10 carrier waves instead of 18 carrier waves. It can be concluded from Figure 7 that the transient states of the proposed 19-level inverter have a fast response

with the change in the load and modulation index. Figure 8 shows the voltage of switches and diodes applied to the proposed 19-level inverter. These voltages present the peak inverse voltage (PIV) of each switch and diode.

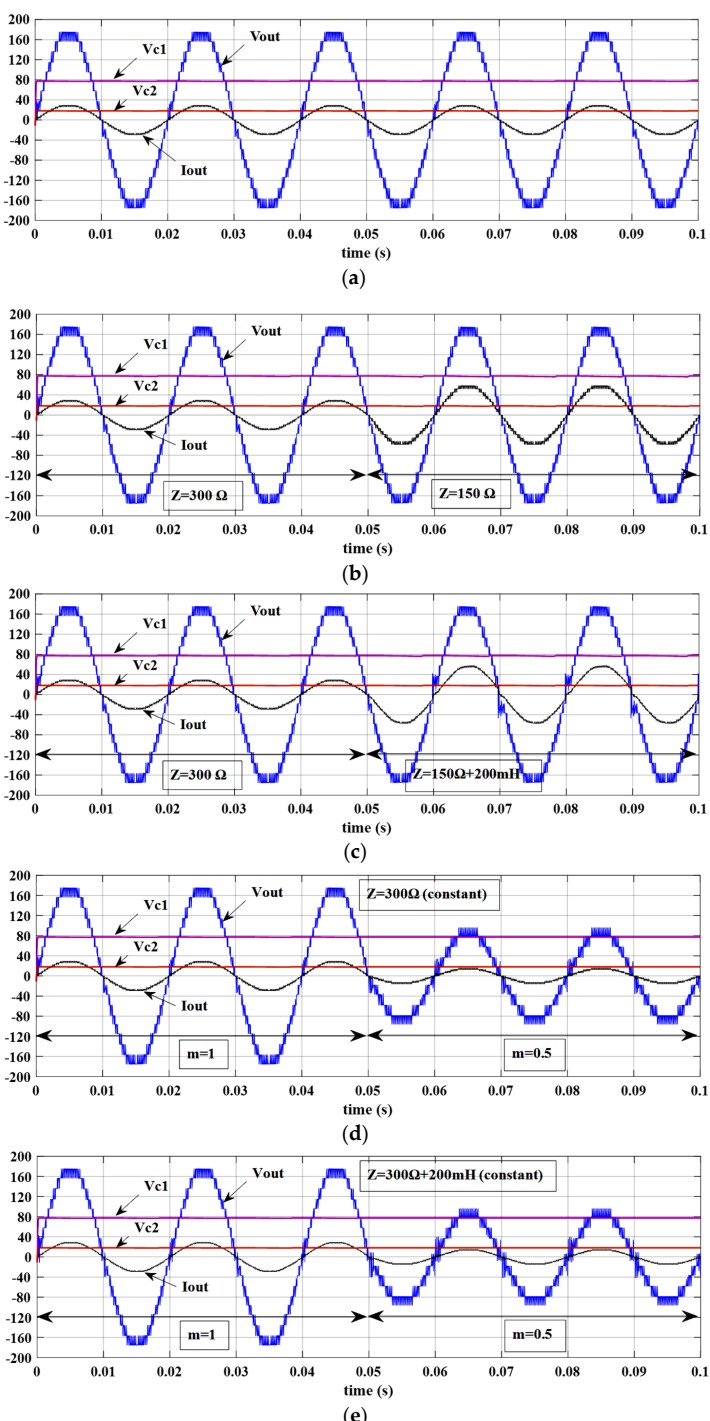

**Figure 7.** Simulation results for output voltage, output current, and capacitor voltages. (**a**) With a constant pure resistive load of Z = 300 Ω. (**b**) With a change in resistive load from Z = 300 Ω to Z = 150 Ω at t = 0.05 s. (**c**) With a change in impedance load from Z = 300 Ω to Z = 150 Ω + 200 mH at t = 0.05 s. (**d**) With a change in modulation index from m = 1 to m = 0.5 at t = 0.05 s under constant pure resistive load Z = 300 Ω. (**e**) With a change in modulation index from m = 1 to m = 0.5 at t = 0.05 s under constant impedance load Z = 300 Ω + 200 mH. In all above figures, for more clarity, the amplitude of output current has been multiplied by 50.

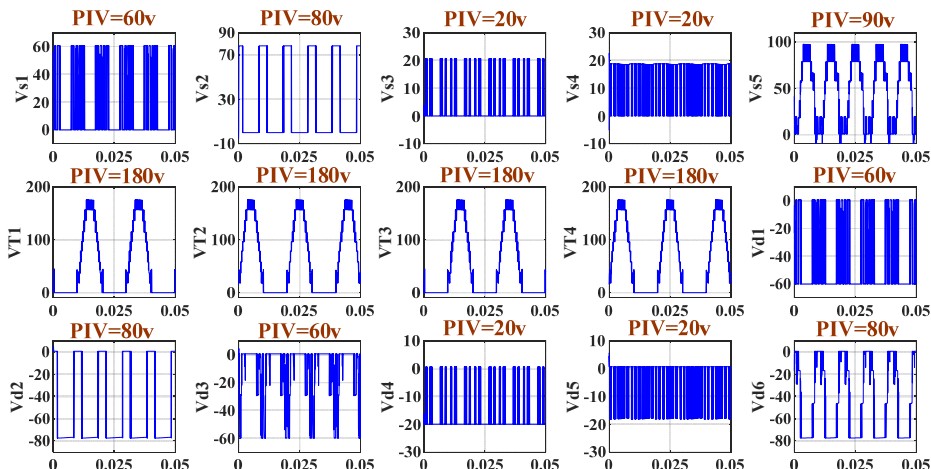

**Figure 8.** Simulation results for PIV of all semiconductor devices applied to the proposed 19-level inverter.

It can be seen from Figure 8 that, among the semiconductor devices, the switches and diodes, $S_3$, $S_4$, $d_4$, and $d_5$, have lower stress voltages than the others because they are arranged inside the small DC bus voltage $u_2 = 20$ V (see Figure 1). According to Figure 8, the sum of the PIV of switches and diodes is equal to 1310, which is called the TSV. By dividing the TSV by the maximum voltage level on the load (180 v), the $TSV^{pu}$ is obtained as 1310/180 = 7.2, which complies with Table 3.

## 7.2. Experimental Results

To assess the simulation results, an experimental setup is implemented using the TMS320F28379D DSP. Figure 9 shows the prototype setup, which includes a DSP, a gate driver, the proposed 19-level inverter, several power supplies, and different resistive-inductive loads. In the gate driver circuit, the HCPL-3120 is used both as a DSP ground isolator and as a switch driver; this needs a power supply of +15 V with the ground ($G_D$). In addition, in the gate driver circuit, the 74HC245 buffer is applied to prevent current consumption by the DSP. In this setup, the switches and diodes are the FGA25N120 IGBT and MBRF20100CT SCHOTTKY diode, respectively. However, within the laboratory environment, it is better to use a low-voltage drop switch such as the STGW50HF60SD IGBT due to limitations in the ranges of available DC voltage sources.

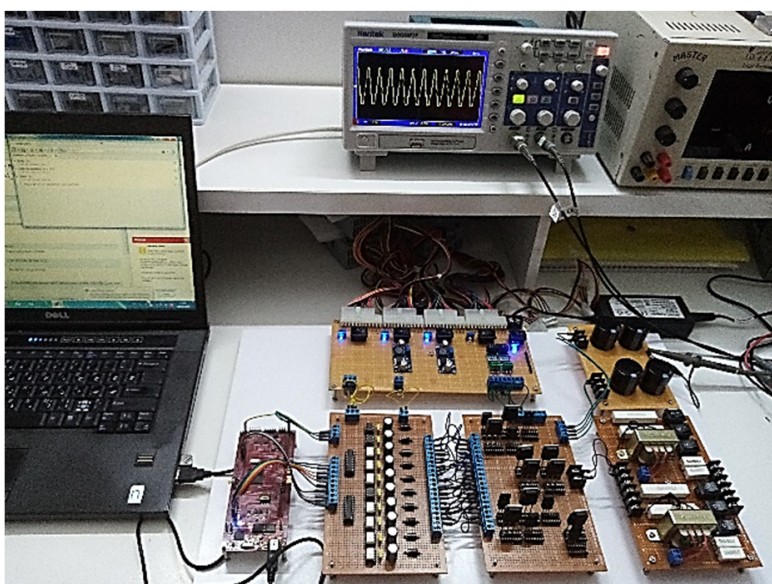

**Figure 9.** Experimental test setup of the 19-level inverter.

Generally, due to the technical limitations encountered in implementations, in the context of our multilevel inverter, a few important points need to be stated:

(1) All gate driver power supplies should be isolated from each other.
(2) The reference signal in the MC-PWM (see Figure 5) must be set on the sample base mode with 1000/5 samples per period and a sample time of 0.0001 s.
(3) A high-power resistor should be applied parallel to each capacitor for discharging their voltage when the test is completed.

The parameters related to the laboratory implementation are listed in Table 4. Figure 10 shows the 19-level output voltage with an amplitude of 150 V. According to the smallest selected voltage level, i.e., 20 V, the maximum voltage level was expected to be 180 volts, but due to the voltage drop across the switches and diodes, this value descends to 150 volts.

**Table 4.** Components of the 19-level inverter in the experimental setup.

| | |
|---|---|
| First input DC-source | $u_1$ = 60 v |
| Second input DC-source | $u_2$ = 20 v |
| Peak output voltage | 180 v |
| Processor | DSP TMS320F28379D |
| Capacitors | $C_1 = C_2$ = 4700 μF |
| IGBT | IRG4IBC30S |
| Diode | MBRF20100CT |
| Driver/optocoupler | HCPL-3120 |
| Current sensor | Resistive divider (0.1 Ω, 40 w) |
| Voltage sensor | Resistive divider (5 × 100 kΩ) |
| Sample time | 10 μs |
| Switching frequency | 5 kHz |
| Output frequency | 50 Hz |
| Resistive load | R = 300 Ω, 150 Ω |
| Resistive-Inductive load | R = 300 Ω, L = 22 mH |

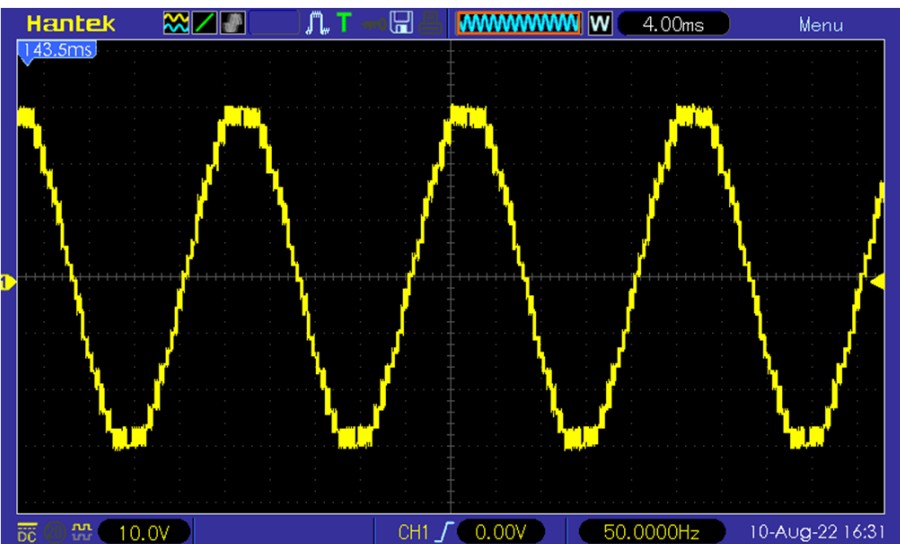

**Figure 10.** Output voltage of the 19-level inverter. In order to obtain the actual values, the vertical axis must be multiplied by a factor of 5.

Figure 11 depicts the output voltage and current of the 19-level inverter when the resistive load changes from Z = 300 Ω to Z = 150 Ω. As shown in Figure 11, the amplitude of the current changes from I = 0.4 A to I = 1 A. Figure 12 shows the output voltage and current when the load changes from pure resistive Z = 300 Ω to Z = 300 Ω + 22 mH. According to Figure 12, the current approximately mimics a sinusoidal waveform with a peak of 0.4 A. Figure 13 depicts the voltages of the capacitors. According to this figure, capacitors

$C_1$ and $C_2$ are charged up to about 75 V and 16 V, respectively. Given the values of the DC sources as $u_1$ = 60 v and $u_2$ = 20 v, capacitors $C_1$ and $C_2$ were expected to charge up to 80 V and 20 V, respectively. The difference is, again, due to the voltage drop across the switches and diodes. Under a pure resistance load, the frequency spectrum of the harmonic curve is depicted in Figure 14. According to this figure, the THD in the output of the inverter is 7.4%, which is less than 8%, complying with the IEEE standards.

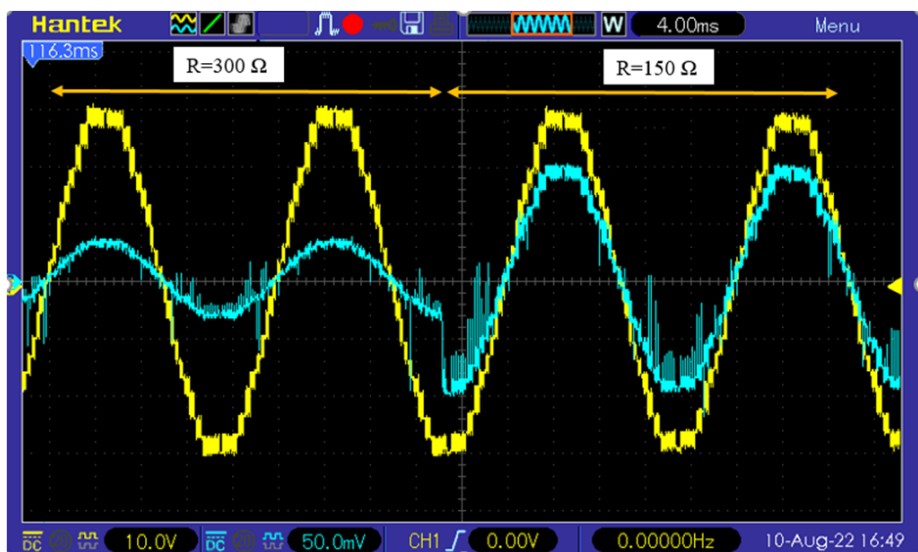

**Figure 11.** Output voltage (yellow wave) and output current (blue wave) of the 19-level inverter when the load changes from R = 300 Ω to R = 150 Ω. In order to obtain the actual values of the corresponding voltage and current, the vertical axis must be multiplied by factors of 5 and 10, respectively (see Table 4).

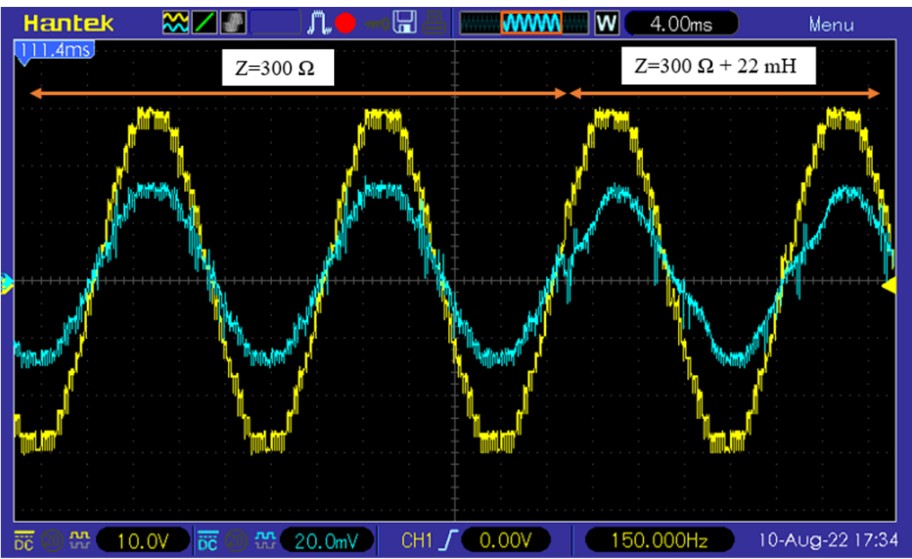

**Figure 12.** Output voltage (yellow wave) and output current (blue wave) of the 19-level inverter when the load changes from Z = 300 Ω to Z = 300 Ω + 22 mH. In order to obtain the actual values of voltage and current, the vertical axis must be multiplied by factors of 5 and 10, respectively.

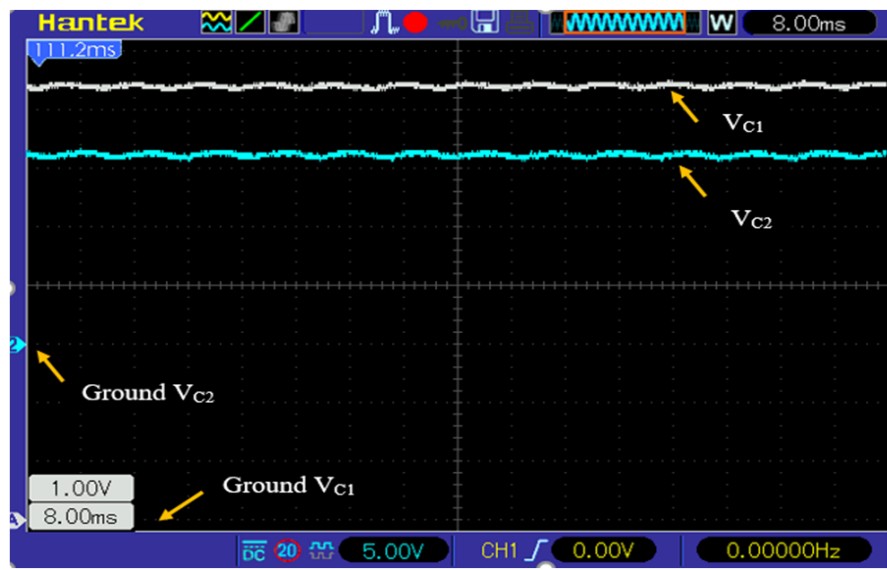

**Figure 13.** Capacitor voltages $C_1$ and $C_2$ (probe oscilloscopes for capacitor voltages $C_1$ and $C_2$ are set on ×10 and ×1, respectively). The values are exact, and there is no need for scaling up/down by a factor.

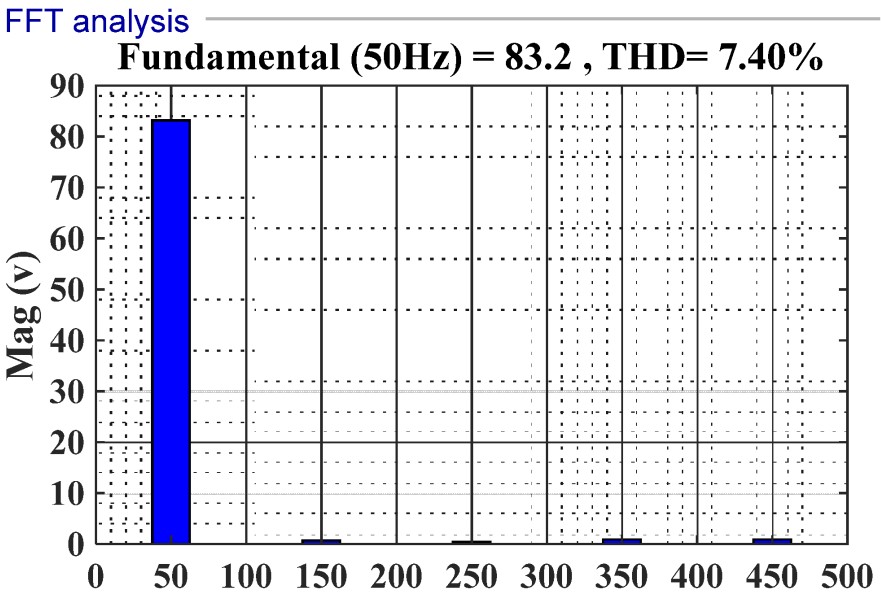

**Figure 14.** Harmonic spectrum curve of output voltage. The vertical and horizontal axes are magnitude (in volts) and frequency (in Hz), respectively.

## 8. Conclusions

In this paper, a 19-level inverter was proposed, which consisted of nine switches, six diodes, two capacitors, and two isolated DC sources. The main advantage of the proposed inverter was the utilization of a very low number of switches and gate drivers compared to other suggested structures. The voltage gain of the proposed inverter was 2.25. The THD of the output voltage achieved 7.4%, which is less than 8%, complying with the IEEE standards. Another advantage of the proposed inverter was the characteristic of modularity, which means it can easily be extended to attain a higher number of voltage levels. The implementation setup of the proposed 19-level topology showed that the charging and discharging states of the topology followed a self-balanced behavior. Due to having a higher TSV in the proposed multilevel inverter than other topologies, the limitation introduced is only the use of switches and diodes with a higher voltage rating. Losses analysis of

the inverter indicated that the efficiency of the proposed converter, when compared with the international standards, is acceptable for this type of converter. Last but not least, the experimental results verified the performance of the proposed topology.

**Author Contributions:** F.S.: conceived and designed the analysis, developed the theory and performed the computations, carried out the experiment, discussed and verified the results, performed the analysis, wrote the manuscript draft, wrote the final manuscript; J.S.: conceived and designed the analysis, discussed and verified the results, performed the analysis, wrote the manuscript draft, wrote the final manuscript; A.K.: discussed and verified the results, wrote the final manuscript. All authors have read and agreed to the published version of the manuscript.

**Funding:** This research received no external funding.

**Conflicts of Interest:** The authors declare no conflict of interest.

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
