# Peer review of "An Asymmetrical 19-Level Inverter with a Reduced Number of Switches and Capacitors"

_electronics, doi:10.3390/electronics12020338_

Round 1

Reviewer 1 Report

Authors proposed in the article an asymmetrical multi-level inverter that produces 19-level output voltages. The proposed development gives a considerable reduction in the number of switches compared to other topologies presented in the literature. The paper is interesting and well organized; however, I consider that the paper will benefit if the authors address within the manuscript the following aspects:

1.    Voltage stress calculations should be conducted.

2.    The steps applied to decide the magnitude of DC sources should be introduced.

3.    In my opinion, the results and discussion are unsatisfactory in presenting the advantages and disadvantages of the proposed topology.  A numerical analysis, that compares the performance of the suggested topology and other topologies, should be conducted. The comparison can be done from viewpoints of the number of DC sources, efficiency, quality of the output voltage, total losses, maximum output voltage, and total Voltage Stress.

4.    The conclusion needs to be reworked. The limitations of the proposed topology should be presented.

Reviewer 2 Report

This paper presents new multi-level inverter topology consisting of less number of semiconductor switches and capacitors. Although the authors provide good simulation and experimental results, the paper need more comparisons and investigations since this topic have been addressed in several publications in  the literature. In order to increase the paper contribution, the authors have to address the following points:

-          A detailed comparison needs to be provided in the end of the introduction part with the most recent publications in this topic considering the total standing voltage stresses, output voltage levels and boosting ratio.

-          The proposed topology should be tested with highly inductive/capacitive load in order to investigate the freewheeling issue. The inductance or capacitance values should be more dominant compared to the resistance part.   

-          Both simulation results and experimental results should be added for the same case.  

Reviewer 3 Report

The paper presents “An Asymmetrical 19-Level Inverter with a Reduced Number of 2 Switches and Capacitors”. Please, verify the following points.

1.       Avoid citing references as [13-17]… instead, highlight the context of each one with this paper…

2.       The main contributions must be clearly presented in the introduction. Please, improve it…

3.       In fig.6, the values of the efficiency were estimated, am I right?

4.       The experimental validation sounds interesting with a very interesting DSP and IGBTs… thank you!

5.       How you implement the carriers in the DSP? Please, clarify since it will be very interesting for potential readers…

6.       The experimental results obtained are interesting with a resistive load and in steady-state and transient-state, however, experimental results with other linear loads (RL) are welcome…

7.       In the sequence of the previous comment, also experimental results with nonlinear load are welcome…

8.       The reference list can be improved with more recent papers…

9.       Additionally, please, discus about the unidirectional operation mode, which can be a limitation of the topology…

10.   Additionally, please clarify possible scenarios of applications since it is not easy to consider scenarios with u1=3V and u2=V…

Round 2

Reviewer 1 Report

The manuscript can be accepted in its current form.

Reviewer 2 Report

The reviewer is satisfied with the revised version of the manuscript. The paper is suitable to be published in its current form.

Thank you